# Oral Sequelae after Head and Neck Radiotherapy: RCT Comparing 3D-Printed Tissue Retraction Devices with Conventional Dental Splints

**DOI:** 10.3390/jcm12082789

**Published:** 2023-04-09

**Authors:** Christopher Herpel, Thomas Held, Christos Labis, Leo Christ, Kristin Lang, Sebastian Regnery, Tanja Eichkorn, Adriane Lentz-Hommertgen, Cornelia Jaekel, Julius Moratin, Karl Semmelmayer, Tracy Thecla Moutsis, Karim Plath, Oliver Ristow, Christian Freudlsperger, Sebastian Adeberg, Jürgen Debus, Peter Rammelsberg, Franz Sebastian Schwindling

**Affiliations:** 1Department of Prosthodontics, Heidelberg University Hospital, 69120 Heidelberg, Germany; 2Department of Radiation Oncology, Heidelberg University Hospital, 69120 Heidelberg, Germany; 3Heidelberg Institute of Radiation Oncology (HIRO), 69120 Heidelberg, Germany; 4National Center for Tumor Diseases (NCT), 69120 Heidelberg, Germany; 5Department of Oral and Maxillofacial Surgery, Heidelberg University Hospital, 69120 Heidelberg, Germany; 6Department of Otorhinolaryngology, University of Heidelberg, 69120 Heidelberg, Germany; 7Department of Radiation Oncology, Marburg Ion-Beam Therapy Center (MIT), Marburg University Hospital, 35043 Marburg, Germany; 8Department of Radiation Oncology, Marburg University Hospital, 35043 Marburg, Germany; 9Clinical Cooperation Unit Radiation Oncology, German Cancer Research Center (DKFZ), 69120 Heidelberg, Germany; 10Heidelberg Ion-Beam Therapy Center (HIT), 69120 Heidelberg, Germany; 11German Cancer Consortium (DKTK), Partner Site Heidelberg, German Cancer Research Center (DKFZ), 69120 Heidelberg, Germany; 12Department of Prosthetic Dentistry, Medical University Innsbruck, 6020 Innsbruck, Austria

**Keywords:** tissue retraction device, 3D-printing, head and neck cancer, radiotherapy

## Abstract

Objectives: To evaluate oral sequelae after head and neck radiotherapy (RT) when using two different types of intraoral appliances. Thermoplastic dental splints (active control) protect against backscattered radiation from dental structures. Semi-individualized, 3D-printed tissue retraction devices (TRDs, study group) additionally spare healthy tissue from irradiation. Materials and Methods: A total of 29 patients with head and neck cancer were enrolled in a randomized controlled pilot trial and allocated to receive TRDs (*n* = 15) or conventional splints (*n* = 14). Saliva quality and quantity (Saliva-Check, GC), taste perception (Taste strips, Burghart-Messtechnik), and oral disability (JFLS-8, OHIP-14, maximum mouth opening) were recorded before and 3 months after RT start. Radiotherapy target volume, modality, total dose, fractionation, and imaging guidance were case-dependent. To evaluate intra-group developments between baseline and follow-up, nonparametric Wilcoxon tests were performed. Mann-Whitney-U tests were applied for inter-group comparisons. Results: At follow-up, taste perception was unimpaired (median difference in the total score; TRDs: 0, control: 0). No significant changes were found regarding oral disability. Saliva quantity (stimulated flow) was significantly reduced with conventional splints (median −4 mL, *p* = 0.016), while it decreased insignificantly with TRDs (median −2 mL, *p* = 0.07). Follow-up was attended by 9/15 study group participants (control 13/14). Inter-group comparisons showed no significant differences but a tendency towards a better outcome for disability and saliva quality in the intervention group. Conclusion: Due to the small cohort size and the heterogeneity of the sample, the results must be interpreted with reservation. Further research must confirm the positive trends of TRD application. Negative side-effects of TRD application seem improbable.

## 1. Introduction

Head and neck cancer is the sixth most common type of cancer, representing about 4–6% of all cancer cases [1,2,3]. Radiotherapy (RT) is used to treat approximately 80% of patients with cancer of the head and neck [4]. The National Comprehensive Cancer Network Clinical Practice Guidelines in Oncology (NCCN Guidelines) have stated “Principles of Dental Evaluation and Management” for head and neck cancers. Among other goals, the effects of radiation backscatter should be minimized in patients with metal dental restorations by the use of thermoplastic splints. Furthermore, a potent topical fluoride should be applied daily. The silicone splints are typically 3 mm thick, reaching up to 5 mm [5,6]. They are worn intraorally during RT—in addition to immobilization masks, which decrease rotation, flexion, and extension of the head. Conventional thermoplastic splints are currently provided for almost every fully and partially dentate patient undergoing RT, and they are commonly used for fluoride application as well [7].

Tissue retraction devices (TRDs) are employed to displace healthy tissue out of the irradiation field and to immobilize the target volume [8,9]. By exposing healthy tissue to less radiation, acute toxicity is reduced in these regions. Acute toxicity is a prominent issue in clinical practice caused by the depletion of rapidly proliferating normal tissue cells. Clinical manifestations comprise radiation-induced oral mucositis, xerostomia, dysgeusia, and trismus [10]. Patients experience pain and difficulties with food intake. In severe cases, parenteral nutrition becomes necessary, or therapies have to be discontinued.

Even small geometric changes resulting from a TRD can substantially reduce radiation exposure in healthy tissue. Although the effectiveness of TRDs has already been demonstrated in smaller retrospective cohorts [11,12,13], most clinical evidence is based on case reports or series [14,15,16,17]. This seems plausible since tissue retraction is highly individual for each patient. Since a TRD is a patient-specific intraoral device, it conventionally requires an elaborate procedure, including dental impressions, cast fabrication and analysis, design, fabrication, and intraoral adjustments. This procedure is difficult to integrate into the clinical workflow of radiation planning [12,14]. Therefore, we developed a semi-customized 3D-printed splint system, which can be adapted to an individual patient within a single 30-min-long dental appointment. The feasibility and patient compliance with TRDs, their geometric reproducibility throughout the RT period, as well as their effects on irradiation planning were previously demonstrated [18].

However, a direct comparison of conventional dental splints and TRDs regarding acute oral sequelae after RT is yet missing. Apart from radio-induced mucositis, which will be the subject of a separate manuscript, the following plausible effects of tissue retraction have been suggested: (i) It has been hypothesized that sparing healthy tissue could mitigate radiation-induced *xerostomia* if major or minor salivary glands, which are located throughout the oral cavity, receive less radiation dose due to tissue retraction [9,19,20]. Xerostomia is considered the main cause of radiation caries and occurs with a prevalence of 30–60% after radiation. (ii) Moreover, it seems conceivable that tissue retraction would reduce *dysgeusia* after irradiation [21]. Dysgeusia usually occurs in such cases because taste buds are affected by irradiation. (iii) Finally, a disease- and treatment-specific *disability* of patients is to be expected in general. More specifically, reduced mouth opening (trismus) was frequently reported after head and neck irradiation. TRD use could be protective against trismus due to a possible “training function” of regularly enforced mouth opening [12,19,22,23].

The primary objective of the present article is to describe the development of key oral functions and to explore a possible effect of tissue retraction applied in the intervention group. For this purpose, participants underwent dental examinations before treatment and 3 months after radiotherapy started. The following null hypothesis was tested: Conventional splints and TRDs perform equally with regard to (i) xerostomia, (ii) dysgeusia, and (iii) disability. Alongside this, step-by-step instructions for TRD production (3D printing) and adaptation are provided to enable further collaborations between surgeons, dentists and radiotherapists.

## 2. Materials and Methods

### 2.1. Study Design

The GUARD study was designed as a non-blinded, 1:1 parallel-group randomized controlled pilot trial and approved by the local ethics committee. An a priori study protocol was pre-registered under ClinicalTrials.gov, NCT04454697, GUARD trial, on 1 July 2020 and also published [24]. A feasibility study was also published, which was carried out in preparation for the GUARD study [18]. The study was conducted at Heidelberg University Hospital in collaboration with the Departments of Radiation Oncology, Oral and Maxillofacial Surgery, Otorhinolaryngology, and Prosthodontics. Patients were screened for eligibility and recruited between 8/2020 and 12/2021.

Inclusion criteria were (i) diagnosis of malignant tumor with a clinical target volume within the oral cavity or adjacent structures; (ii) indication for adjuvant or definitive radiotherapy; (iii) a minimum age of 18 years; (iv) a Karnovsky score of at least 60; (v) completed wound healing after the surgical intervention; (vi) adequate contraception for women and men of childbearing age and (vii) the ability of to give informed consent. Exclusion criteria were (i) previous head and neck radiotherapy; (ii) multifocal tumor growth; (ii) unrecovered condition as a result of acute toxicity of previous treatments; (iii) mouth opening less than 2 cm and (iv) simultaneous chemotherapy.

Informed consent was obtained from all eligible patients willing to participate. Enrolled participants were randomly assigned into two groups in a 1:1 ratio. They received either a TRD (intervention) or a thermoplastic dental splint for each jaw (active control). The outcome measures were collected at baseline and 3 months after radiotherapy started. Radiation therapy was planned individually for each participant and was not study-specific. The definition of the target volume and dose prescription was based on current clinical guidelines in each case. In addition to their splints/TRD, the usual thermoplastic immobilization mask was worn during irradiation sessions. Figure 1 shows exemplary patients to illustrate the effect of tissue retraction on radiation planning.

### 2.2. Active-Control Group: Dental Splints

Participants in the control group received an active control consisting of soft, flexible dental splints with a layer thickness of 5 mm for the maxilla and mandible. These splints are the standard treatment for patients with radiotherapy of head and neck cancer according to NCCN guidelines [25]. Their rationale is based on studies describing a backscatter effect of dental materials. The shielding achieved by the splints is meant to protect the adjacent soft tissue [26,27]. For this purpose, alginate impressions of the maxilla and mandible were taken, and the splints were fabricated (material: Erkoflex 5 mm, Erkodent, Pfalzgrafenweiler, Germany; thermoforming device: Erkoform RVE, Erkodent, Figure 2) by thermoforming on gypsum casts.

### 2.3. Intervention Group: Tissue Retracting Device

The semi-individualized TRDs were intraorally adjusted by one of two investigators. The technique was described elsewhere [18] but was optimized for this study on the basis of the following considerations: (i) first, the devices should integrate the protection against backscatter radiation provided by the splints used in the control group. (ii) Second, it should be considered that tissue retraction would involve the retraction of soft tissues of individual anatomy and mobility, which spoke against a fully digital (based on image data) or fully laboratory (based on gypsum casts) design. (iii) Third, the method should be integrated into the clinical routine in a time-efficient manner without risking a delay in the treatment schedule, which militated against a fully individualized solution or one that required multiple appointments.

Therefore, a semi-customized splint system was developed in which a 3D-printed prefabricated splint is individualized within a single 30-min-long dental visit. Since the intraoral anatomical relations are rather individual, a limited number of ready-made TRD sizes must be adaptable to different patients and therapy modalities. The final TRD design was based on the average proportions of the dental arches, temporomandibular joints and surrounding hard and soft tissues. The design was developed using multiple prototypes and tested on volunteers. Specific design changes were made to facilitate the insertion and adaptation of the TRD and to prevent pressure points or triggering of the gag reflex.

Figure 3 shows the workflow for the production and adaptation of the developed TRD. Printing was done in a layer thickness of 100 µm (printer: Asiga MAX, printing resin: V-Print splint, VOCO, Cuxhaven, Germany). After the removal of the support structures, post-processing was carried out by washing in an isopropanol bath, drying with compressed air and curing by means of 2 × 2000 xenon-light flashes (Otoflash G171, NK-Optik, Baierbrunn, Germany). The devices were printed in three different sizes (small, medium and large, Figure 3a) and kept in stock. For an individual patient, the appropriate size was first selected and tried on (Figure 3b,c). By removing parts of the splint, it was then adjusted to avoid painful pressing on the mucosa at any point and to avoid triggering the gag reflex (Figure 3d). After the adhesive was applied (Figure 3e), the top and bottom of the splint were filled with silicone impression material (Flexitime Dynamix Putty, Kulzer, Hanau, Germany; Figure 3f). The impression was taken simultaneously in both jaws by first adapting the splint to the maxillary teeth with opened mouth and then asking the patient to bite into the silicone with a slight protrusion. After removal, excess impression material was cut away with a scalpel (Figure 3g,h). Patients were instructed on how to insert and remove the splints, respectively, and the TRD for their radiation sessions.

To make it easier for the interested reader to reproduce the described method of tissue retraction in their own clinical setting, the STL construction files are available under a CC-BY license from the corresponding author upon reasonable request.

### 2.4. Outcome Measures and Statistical Analyses

#### 2.4.1. Xerostomia Assessment

Saliva quality testing was performed using an established test kit (Saliva-Check BUFFER kit, GC) [28]. Reduced-stimulated and unstimulated saliva production and increased viscosity are markers of xerostomia and are often accompanied by a reduced pH value and buffering capacity. The used test consisted of five assays, three for unstimulated saliva and two for stimulated saliva: (i) Resting flow rate could take three values (0, 1 and 2; corresponding to low, normal, and high flow rates) and was a measure of unstimulated saliva production. It was defined as the time required for the lower labial glands to produce visible saliva, with a normal flow rate ranging from 30–60 s. (ii) Saliva consistency was also visually assessed on a three-point scale from 0–2, thereby classified as “severely increased viscosity,” “increased viscosity,” and “normal viscosity,” which was accepted for water-clear saliva. (iii) For the measurement of pH, participants were asked to expectorate into a cup. The pH was then measured with an indicator paper. A healthy saliva pH value is assumed between 6.8–7.8. (iv) To measure the stimulated saliva production, the participants chewed a chewing gum for 30 s and then expectorated any saliva that accumulated during the following 5 min of continued chewing. A value of >5 mL corresponded to a normal stimulated saliva flow rate. (v) Finally, the buffering capacity was measured in the collected saliva using indicator paper. It took values between 0 and 12, with 10–12 being considered as normal buffering capacity.

#### 2.4.2. Dysgeusia Assessment

In order to objectively assess the impairment of the sense of taste, a validated gustatory test was carried out (Taste Strips, Burghart Messtechnik, Holm, Germany). In this test, 18 test strips of filter paper were used, each of which carried one of four taste qualities (sweet, sour, salty, bitter) in one of four concentrations. The strips were presented in a pseudo-randomized order to the participants, who were asked to identify the respective taste (“sweet,” “sour”, “salty”, or “bitter”). To obtain a total score, each correct answer was counted as one point. The set contained 2 control strips, which did not contain any taste and were not counted. Therefore, a total score of between 0 and 16 could be achieved [29].

#### 2.4.3. Disability Assessment

Participants were questioned at baseline and three-month after the start of radiotherapy using the Oral Health Impact Profile (OHIP) and the Jaw Functional Impairment Scale (JFLS). Both are commonly used questionnaires for the measurement of disability. While OHIP represents a more generic, complex construct, JFLS addresses the condition-specific, functional disability [30].

OHIP-14: The 14-item short version of the Oral Health Impact Profile was used. The instrument is designed as a comprehensive measure of oral health-related quality of life [31,32]. Participants reported their experience of various oral health-related impairments on 5-point scales ranging from 0 (never occurring) to 4 (very often). A sum score (range 0–56) was calculated from all items, with a higher score indicating greater impairment of individual well-being and a higher social impact due to oral conditions. Standard values have been determined for the German general population [33]. The validated German version showed high internal reliability (Cronbach’s α = 0.90) [34].

JFLS-8: The Jaw functional limitation scale was used in its 8-item version to assess the functional impairment of participants. The JFLS was developed and validated for different diagnostic groups to assess the functional status of the masticatory system and showed good internal reliability in previous studies (Cronbach’s α = 0.87) [35]. Participants indicated the severity of their limitation using 8 items on a 10-point numerical rating scale. A global score (range 0–10) was calculated as the mean of all items answered. A higher score indicated greater limitation in masticatory function, jaw mobility, and emotional and verbal communication. The JFLS was used in its German translation [36]. As an additional marker of possible radiation-induced trismus, the participants’ maximum voluntary mouth opening was measured as maximum inter-incisal distance.

#### 2.4.4. Statistical Analysis

For the GUARD study, sample size calculation was performed for the incidence of oral mucositis. It was based on preliminary clinical experience with TRDs and was published as part of the study protocol [24]. A necessary number of 28 evaluable participants (14 per group) was determined to achieve a power of 80% at a significance level of 5%.

The statistical analysis was performed using the software SPSS Statistics 25 (IBM). Outcome variables were analyzed using descriptive methods and assessed for normal distribution using the Shapiro-Wilk test. Effects of radiotherapy were examined using the Wilcoxon signed rank test. A potential effect of the intervention (tissue retraction group) was analyzed by group comparison using the Mann-Whitney U test. As no sample size estimation was performed for the long-term oral outcomes reported in this paper, the analysis must be considered exploratory, and all *p*-values were reported exactly and regarded as continuous parameters reflecting the level of evidence.

## 3. Results

### 3.1. Study Recruitment and Execution

Figure 4 shows the flowchart of study participation. Among the patients who met the inclusion criteria, 29 consented to participate in the study and were randomized. All of them received either a tissue retraction device (15) or dental splints (14) prior to radiotherapy treatment planning. Participants were enrolled between August 2020 and December 2021 until the recruitment target was reached. All participants who received a TRD completed radiotherapy.

To assess oral sequelae after RT, 13 of 14 participants in the control group and 9 of 15 participants in the intervention group were clinically reexamined. All participants who were reexamined were included in the analysis. Their baseline characteristics are presented in Table 1.

### 3.2. Oral Sequelae

It was analyzed whether the assessed oral variables would differ between baseline and three months after the start of radiotherapy. Table 2 (left part) shows the medians of the changes for both the active control and intervention group. Positive values indicate an increase and negative values are a decrease compared to the baseline. The significance of the differences was assessed by means of the two-tailed Wilcoxon signed rank test after the non-normal distribution was determined with the Shapiro-Wilk test. Assuming an alpha level of 0.05, there was evidence for the development of xerostomia in the active control group with moderate to strong effect sizes. The resting flow rate and the stimulated saliva quantity decreased (r = 0.62, *p* = 0.025/r = 0.67, *p* = 0.016), and salivary consistency worsened (r = 0.57/*p* = 0.038). In contrast, no significant changes were found in the intervention group. In both groups, no significant changes were found for the other two markers of saliva quality (pH resting saliva and buffering capacity stimulated saliva) and for the outcomes of dysgeusia and disability.

### 3.3. Comparison Intervention and Control Group

The changes in the outcome variables (difference between 3-month follow-up and baseline measurement) were subsequently examined for differences between the intervention group and the control group using the Mann-Whitney-U test (Table 2, right column). Hereby, a possible positive influence of tissue retraction should be investigated. Effect sizes were consistently small to medium, and the differences did not reach a 0.05 significance threshold.

## 4. Discussion

The purpose of this article was to describe the oral sequelae observed within the GUARD study and to examine the influence of TRDs on these sequelae. The concept of tissue retraction presented here was found to be non-inferior to the standard therapy (active control) using dental splints. Rather, there was a tendency for the intervention group to perform better in terms of xerostomia and maximum mouth opening. A significant deterioration of saliva quality was found in the active control group but not in the intervention group. In the group comparison, however, these trends did not reach a significance threshold of 0.05.

The impairment of stimulated and unstimulated saliva secretion observed in both groups is consistent with previous studies describing permanent damage to saliva production after irradiation [28,37,38]. Moller et al., who measured several time points, described an improvement in saliva production three months after radiotherapy, which, however, did not result in a complete recovery [39]. Furthermore, for buffering capacity, a permanent deterioration was observed even 12 months after radiotherapy [39]. Contrary to this, and in agreement with our results, other studies have shown that a recovery of the capacity can already be observed three months after the start of radiotherapy, and thus only a temporary impairment would be expected [28].

Regarding dysgeusia, no significant differences were found in the present study between pre- and post-irradiation. Also, in this regard, there are partly contradictory reports in the literature. For example, a complete recovery of taste function between the first and third month after the end of treatment was described in a study that used the same measuring strips as the present investigation. [40]. In another study, a return to baseline taste performance was reported not before 6 months after RT. [41]. In contrast, other trials found only incomplete recovery of the sense of taste even after longer follow-up intervals [42,43].

In terms of disability, the measures of oral health-related quality of life, functional jaw impairment and maximum mouth opening were surveyed, and a tendency to deterioration was observed in each case. The fact that the treatment of head and neck cancer leads to a certain degree of disability has already been shown in previous studies on quality of life and global functionality. [44,45]. The more specific jaw functional impairment is manifested, amongst other symptoms, by a reduced mouth opening (trismus) as a result of radiation-induced fibrosis of the masticatory muscles. An incidence of 42% was described in this context after treatment of head and neck tumors, whereby a high variability was reported depending on tumor location, initial tumor size and treatment regime [46].

In this randomized trial, a tendency was found for a better performance of the intervention group compared to the active control group. Moderate effect sizes were found for the variables of saliva testing and maximum mouth opening, while the differences were found to be non-significant at an alpha level of 0.05. Thus, the null hypothesis could not be rejected. However, as no sample size calculation had been done for the investigated outcome measures, the study may have been underpowered to prove the effect of tissue retraction. Previously, Mall et al. and Goel et al. were able to show in controlled studies that TRDs contribute to the prevention of xerostomia [22,47].

In these studies, however, only patients with tongue cancer were included. In addition, intensity-modulated radiotherapy (IMRT) was not applied in any of these studies. After all, IMRT has been shown to reduce xerostomia as a complication of radiotherapy and to generally improve QoL [48,49]. It is also possible that a pronounced inter-group difference was not detected in the present study, as the follow-up examination was performed too late, or vice versa: An advantage of the TRDs under ongoing RT might have been missed in our study.

Furthermore, in the present study, an active control (dental splints) was favored over a no-treatment control group. Even though the dental splints did not allow for tongue displacement, they still contained a certain amount of cheek spacing and increased mouth opening, which possibly already had a therapeutic effect compared to irradiation without any intraoral device at all.

In addition to the investigation of oral sequelae, the present manuscript aims to describe the technical and clinical procedure for the application of a semi-individualized concept of tissue retraction. The developed concept proved to be reliable. All randomized participants were successfully treated with a TRD and underwent radiotherapy as planned. No modifications of either any retractor or any irradiation plan were necessary during treatment. As a result of this positive experience, TRDs were integrated as a regular part of patient care at Heidelberg University Hospital and are being deployed in collaboration between the departments for radiation oncology, maxillofacial surgery, and prosthodontics.

The main limitation of this study is certainly the small sample size and the fact that no sample calculation was performed for the variables investigated in this manuscript. Therefore, the results can only be evaluated exploratively, and it remains to be seen whether the observed small positive tendencies of tissue retraction on saliva quality and maximum mouth opening can be confirmed. Furthermore, the investigated sample was characterized by a certain heterogeneity with regard to the tumor site. More extensive studies with larger sample sizes could show whether there are differences in the effect of tissue retraction depending on the target volume, i.e., whether certain patient groups would particularly benefit from a TRD while others would not benefit at all. Finally, only a single follow-up was conducted after three months. Studies with repeated measurements at multiple time points could better reflect dynamic changes, such as the recovery of saliva production or taste function.

## 5. Conclusions

The presented concept for semi-individualized tissue retraction using prefabricated 3D-printed splints proved to be well feasible and showed complete adherence of the participants treated. There was a small tendency for a positive effect regarding saliva quality and maximum mouth opening three months after the start of radiotherapy compared to the standard treatment with dental splints. However, the differences were below a significance threshold of 0.05. More comprehensive studies with larger samples, which would also allow subgroup analyses, are required to confirm or reject the observed tendencies. Data from the GUARD study regarding the primary outcome of radio-induced mucositis will be the subject of a separate manuscript.

## Figures and Tables

**Figure 1 jcm-12-02789-f001:**
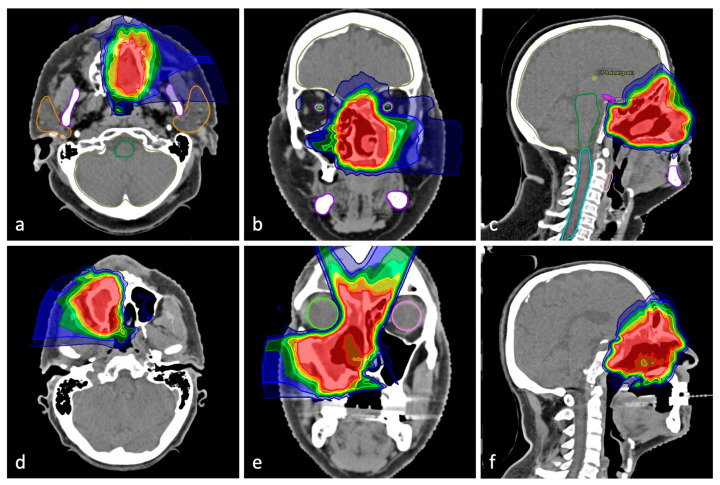
Sectional images of an exemplary patient from the control group (**a**–**c**); melanoma of the nasal cavity) and from the intervention group (**d**–**f**); sinonasal undifferentiated carcinoma). When comparing the radiation treatment plans in coronal and sagittal sections, it can be observed how tissue retraction results in a significant dose reduction in the area of the tongue. Red color indicates areas with a high radiation dose, blue color those with a lower dose.

**Figure 2 jcm-12-02789-f002:**
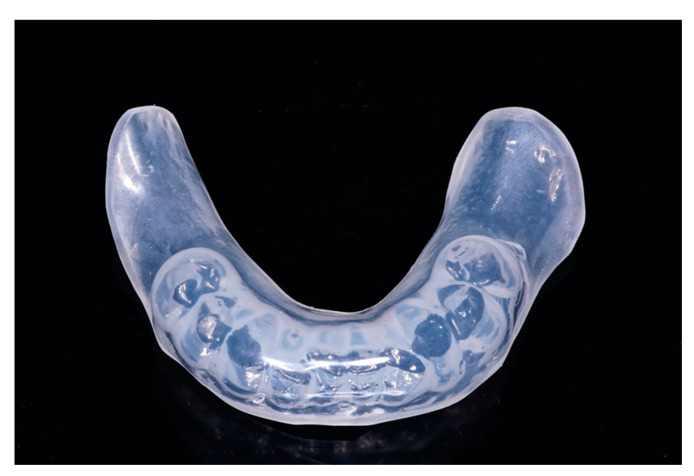
Example of a dental splint that patients in the active control group received in pairs for the upper and lower jaw. In this case, the 5 mm thick thermoplastic splint covers the residual mandibular dentition from tooth 35 to 45.

**Figure 3 jcm-12-02789-f003:**
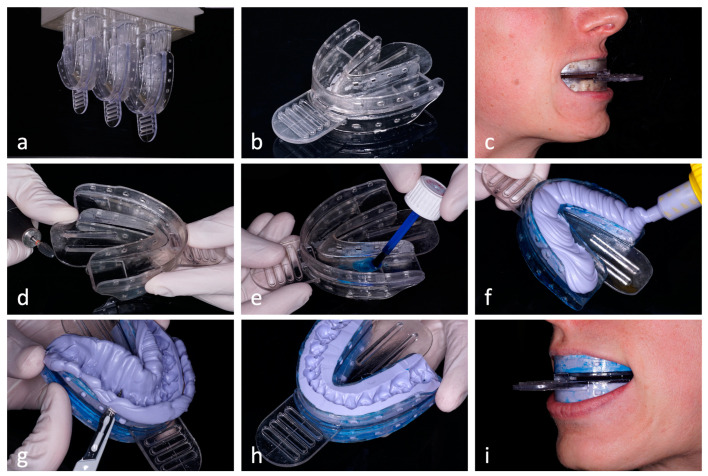
The manufacturing and clinical adaptation workflow for the TRDs is shown. TRDs are printed in three different sizes and kept in stock (**a**). The most suitable size is selected for an individual patient (**b**,**c**). Adjustments are made to avoid the gag reflex and pressure sores (**d**). After the adhesive has been applied (**e**), both upper and lower sides are filled with silicone impression material and placed intraorally (**f**). After the material is set, the excess is removed (**g**,**h**), and the correct fit on the patient is checked (**i**).

**Figure 4 jcm-12-02789-f004:**
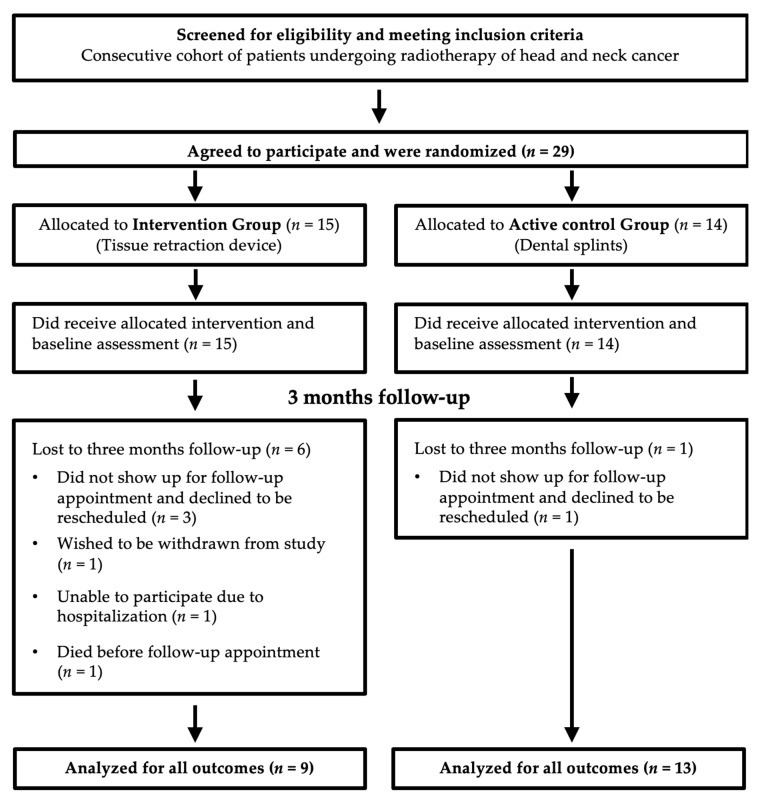
Flow diagram of study participation.

**Table 1 jcm-12-02789-t001:** Baseline characteristics of analyzed patients in the intervention group (tissue retracting device) and active control group (dental splints).

Parameter	Intervention Group (*n* Patients = 9) Median (Range) or Count (%)	Control Group (*n* Patients = 13) Median (Range) or Count (%)
Age	73 (53–83)	66 (41–76)
Sex		
Female	3 (33)	5 (39)
Male	6 (67)	8 (61)
Tumor sites		
Nasal and paranasal sinus	2 (22)	6 (46)
Tongue or mouth base	3 (33)	3 (23)
Maxilla and palate	2 (22)	3 (23)
Mandibula	1 (11)	-
Tonsils	1 (11)	-
Cheek	-	1 (8)
Tumor stage *		
Carcinoma in situ	-	1 (8)
T2	1 (11)	2(15)
T3	2 (22)	5 (39)
T4	3 (33)	5 (39)
TX	1 (11)	-
CUP	2 (22)	-
Tumor histology		
Adenocarcinoma	-	2
Squamous cell carcinoma	4	3
Adenoid cystic carcinoma	2	2
Mucoepidermoid carcinoma	-	1
Others	3	5
Tumor surgery prior to radiotherapy		
Biopsy	3	-
Resection	6	12
None	-	1
Treatment concept		
Definitive	3	1
Adjuvant	5	7
Additive	1	5
Radiation doses to tongue: Mean (SD) **		
Dmean (GyE)	33.6 (23.7)	38.5 (12.8)

* According to the 8th edition of the Union for International Cancer Control (UICC) tumor node metastasis (TNM) system; TX: Main tumor cannot be measured; CUP: Carcinoma of Unknown Primary; ** Difference between mean radiation doses was non-significant on an alpha level of 0.05 (*p* = 0.56; two-tailed unpaired *t*-test); Dmean: mean dose.

**Table 2 jcm-12-02789-t002:** The left part of the table shows the median changes in the outcome variables after three months, separately for the intervention and control groups. Negative values indicate a decrease, and positive values are an increase compared to the baseline measurements. Quartiles are shown in brackets. The significance of the changes was tested using a two-tailed Wilcoxon signed rank test. In the right part of the table (group comparison), the Mann-Whitney U test was used to test whether the observed changes T0–T3 would differ between the intervention and control groups. Exact *p*-values are given for all tests. Values below 0.05 are in bold. Effect sizes are reported using Pearson’s r.

	Median Changes 3 Months after RT (T0−T3)						Group Comparison	
Variables	Intervention Group (*n* = 9)			Control Group (*n* = 13)			*p* Value	Effect Size ^®^
	Median (Quartiles)	*p* Value	Effect Size ^®^	Median (Quartiles)	*p* Value	Effect Size ^®^		
** *Xerostomia* **								
Resting flow rate	0 (−0.5, 0)	0.16	0.47	0 (−1, 0)	0.025 *	0.62	0.56	0.17
Salivary consistency	0 (−0.5, 0)	0.16	0.47	0 (−1, 0)	0.038 *	0.57	0.47	0.20
pH resting saliva	0 (−0.3, 0.4)	0.87	0.06	−0.2 (−0.8, 0.1)	0.064	0.50	0.16	0.30
Quantity stimulated saliva	−2 (−3.8, 1)	0.07	0.61	−4 (−7.5, −0.5)	0.016	0.67	0.32	0.22
Buffering stimulated saliva	0 (0, 2)	0.46	0.25	0 (−3, 0)	0.2	0.35	0.19	0.31
** *Dysgeusia* **								
Total score	0 (−3.5, 20.5)	0.73	0.12	0 (−2, 3)	1	0.0	0.86	0.05
** *Oral disability* **								
OHIP−14	2 (−0.5, 3)	0.14	0.49	1 (−1, 12)	0.24	0.32	0.95	0.02
JFLS−8	−1 (−11, 9)	1	0.0	1 (0, 4)	0.16	0.39	0.65	0.11
Maximum mouth opening	−1 (−3, 2.5)	0.47	0.24	−3 (−4, 0)	0.26	0.31	0.43	0.18

* Please note that significant differences in rank sums after three months were found for both Resting flow rate and Salivary consistency, although median changes were 0 for both variables. *p* values below an alpha level of 0.05 are marked in bold.

## Data Availability

The data presented in this study are available on request from the corresponding author.

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
