# Peer review of "Oral Sequelae after Head and Neck Radiotherapy: RCT Comparing 3D-Printed Tissue Retraction Devices with Conventional Dental Splints"

_jcm, 2023, doi:10.3390/jcm12082789_

Round 1
Reviewer 1 Report
This article concept is very interesting.
But as authors described this article sample size is small and there were no clear results were appeared. More patient’s date should be added.Results
L. 173-188: Change Figure 1 to 2.
Table
Table of radiation doses of tongue is needed for present the difference between two groups.
Figures
Please add the figures of dental splint of active control group. Because difficulty for understanding difference in the present form.
Reviewer 2 Report
The authors present secondary data from a randomized clinical trial using two different types of intraoral tissue retraction devices during radiotherapy. Objective data like saliva quantity and interincisal distance as well as patient-reported outcomes have been collected. The study was approved by the local ethic committee and pre-registered. The use of the English language and wording is well chosen. The title reflects the content. No important references are missing. The differences found between both devices are small. However, the power of the study is low too. Due to the limited available data in the literature regarding this topic, there is still value in the data presented. Special mention is the intention of the authors to share the STL-files of the devices with the community. There are no major flaws within the manuscript.
Results:
Figure 3:
Typo: “follow-up (n=5)” should be “(n=6)”
These sentences repeat the content of figure 3: „One participant died before the follow-up and one patient could not participate due to hospitalization. One participant wished to be withdrawn from study without giving any reasons and four other patients missed their follow-up appointment without excuse and did not want to be rescheduled because of the long journey to the study center.“ Remove the duplicity.
Table 1:
Explaining the abbreviations would be great, especially TX and CUP. Furthermore, the classification applied should be mentioned.
Table 2:
Typo „-2 (-3,8, 1)“
You write: „The median changes in the outcome variables after three months are shown both for the total sample of the GUARD study (left column) and separately for intervention and control groups.“ However, in the table, no total sample data are presented.
Resting flow rate and salivary consistency are looking suspiciously similar. Please check.
This table is overall hard to read. It would be helpful for better guidance to include „T0-T3“ in the header and provide the scales of the variables. Because there is no comparison between the intervention and the control group, separating the tables physical or optical may be helpful. I am not sure why the authors choose to provide median values instead of mean values + SD. Reporting a significant difference and presenting a median change of „0“ in both groups is not helpful.
Basic data from table 2 are presented in table 3 too. Maybe an option is to move table 2 into the annex to make the manuscript more straightforward.
Reporting tendencies without any statistical proof is difficult, especially in the results section. The results section should provide facts only, but no speculative interpretation.
Discussion:
You state: “Rather, there was a consistent tendency for the intervention group to perform better in terms of saliva quality and oral disability.” That is wrong. The intervention group worsened in the median by 2 OHIP points, while the control group only by 1 OHIP point (see table 3).
Round 2
Reviewer 1 Report
Thank you for revising your article.
I think that this article is acceptable.